# Stable heritability of type 1 diabetes in a Swedish Nationwide Cohort Study

Yuxia Wei [1] ✉, Tomas Andersson[1], Shengxin Liu[2], Maria Feychting [1],
Ralf Kuja-Halkola [2] & Sofia Carlsson [1] ✉

Incidence of type 1 diabetes is increasing globally, which is hypothesized to be due to environmental influences. We leverage Swedish nationwide registers linked to all children ($n = 2{,}928{,}704$) born in 1982–2010 to investigate if the heritability of childhood-onset type 1 diabetes has changed over time and how alterations in environmental factors have contributed to the rising type 1 diabetes incidence. The heritability is estimated at 0.83 (95% confidence interval: 0.79, 0.86) and stable over the observation period (0.80 [0.71, 0.86] in 1982, 0.83 [0.79, 0.86] in 2000, and 0·83 [0.79, 0.86] in 2010, respectively). Environmental factors including maternal smoking during pregnancy and childhood adiposity explain <10% of the increasing type 1 diabetes incidence. In this work, the heritability of childhood-onset type 1 diabetes has remained high and stable over the last 30 years. Our findings indicate that the available environmental factors are not the major contributors to the rise in type 1 diabetes in Sweden.

The incidence of childhood-onset type 1 diabetes (T1D) has been increasing worldwide since the 1950s, with an annual increase of 3–4%[1,2]. It is unclear if this trend is due to a rise in nongenetic (environmental) risk factors, increasing genetic susceptibility of the population or a combination.

Environmental explanations are generally favored since it seems unlikely that the genetic makeup of the population changes in a few decades and because incidence varies markedly between genetically similar populations[1,3]. Notably, the rise in T1D coincides with a rise in childhood obesity[4], a proposed risk factor for T1D[5,6]. Other suggested risk factors include early-life factors like maternal smoking (protective)[7], higher maternal age at delivery[8] and breastfeeding (protective)[9]. To what extent the prevalence of these factors has changed and contributed to a rise in the incidence of T1D is unclear.

Heritability is the proportion of phenotypic variance (including variance explained by genetic and environmental factors) in a disease explained by all genetic factors. T1D heritability has been estimated at between 50% and 88% in twin or other family-based studies of different populations[10–13] and we recently estimated it at 81% in Swedish children[14]. If the increasing T1D incidence is mainly driven by environmental factors, the relative contribution of environmental factors to

T1D variance would be expected to increase and heritability decrease over time[15]. As far as we know, no previous study has investigated whether T1D heritability has changed over the last decades. However, a previous study from Australia indicates that the proportion of T1D cases with the highest-risk human leukocyte antigen (HLA) genotypes has decreased while the proportion of affected children with intermediate-risk HLA genotypes increased from 1980s to 2000s[16]. Similar findings have been reported from Finland, Sweden, US, and the UK when comparing cases diagnosed 1985-2008 to those diagnosed earlier[17–20]. This indicates that the contribution of HLA genotypes to T1D risk has changed but not lessened over time[16]. Still, only approximately half of the genetic risks of T1D is accounted for by HLA genotypes[16] and a rise in genetic susceptibility could be driven by risk alleles outside of the HLA complex. T1D proposedly has age-related endotypes, with distinct etiology and pathogenesis between individuals diagnosed before the age of seven and those diagnosed after the age of thirteen[21]. The genetic contribution to the etiology of these endotypes is scarcely investigated.

In this work, we set out to clarify if the environmental and genetic influences on childhood-onset T1D (0–18 years) have changed in Sweden over the last 30 years. We analyze the trend of

[1]Institute of Environmental Medicine, Karolinska Institutet, Stockholm, Sweden. [2]Department of Medical Epidemiology and Biostatistics, Karolinska Institutet, Stockholm, Sweden. ✉e-mail: yuxia.wei@ki.se; sofia.carlsson@ki.se

T1D incidence and heritability among three million children born from 1982 to 2010 using nationwide data. We show that the heritability of T1D and its endotypes has remained stable over time. We identify environmental factors associated with T1D and find that changing prevalence of them could only explain a small proportion of the increasing T1D incidence, indicating that they are not the major driver of the rise in T1D.

## Results

Among 2,928,704 children born in 1982–2010, 20,086 (0.7%) developed T1D during follow-up with a median age at diagnosis of 9.9 years. Compared to the total population, children with T1D were more likely to have parents with T1D, be male and large for gestational age and less likely to have mothers who smoked during pregnancy (Supplementary Table 1). T1D incidence increased steadily from the birth year 1982 to 2000, with a cumulative incidence of 0.50% and 0.93% in children born in 1982 and 2000, respectively, and an HR (95% CI) of 1.85 (1.65, 2.08) for birth year 2000 versus 1982 (Fig. 1). The incidence remained stable for children born 2001-2010 (Fig. 1). The increasing trend was observed for all ages but was less obvious in the oldest group (Supplementary Fig. 1).

## Heritability

The heritability of T1D for the whole period was 0.83 (95% CI: 0.79, 0.86; Supplementary Table 2, Fig. 2). Comparing different models, the model (model 1) with moderations of birth year on both the additive genetic and non-shared environment components had the best fit, with β coefficients >0 for moderation on both components (Supplementary Fig. 2; Supplementary Tables 3 and 4). Based on model 1, heritability and the proportion of variance explained by non-shared environment component remained stable at round 0.8 and 0.2 respectively from 1982 to 2010 (Fig. 2). The heritability was 0.80 [95% CI: 0.71, 0.86] for children born in 1982, 0.83 [95% CI: 0.79, 0.86] in 2000, and 0.83 [95% CI: 0.79, 0.86] in 2010, and there was no heritability difference between children born in later years (such as 2000 or 2010) and children born in earlier years (such as 1985, 1990, or 1995; Supplementary Table 5). T1D heritability estimated from models 2, 3 and 4 was also stable over time (Supplementary Fig. 3). Heritability was higher in the younger age groups (0.85 [95% CI: 0.79, 0.90], 0.66 [95% CI: 0.61, 0.72], 0.58 [95% CI: 0.50. 0.66] for age 0-6, 7-12, and 13-18 years, respectively) and remained stable in the different age groups over the whole period (Fig. 3, Supplementary Tables 3–5).

Simulation analysis estimated that T1D heritability would be attenuated to 0.59 if an increase in T1D cumulative incidence by 40

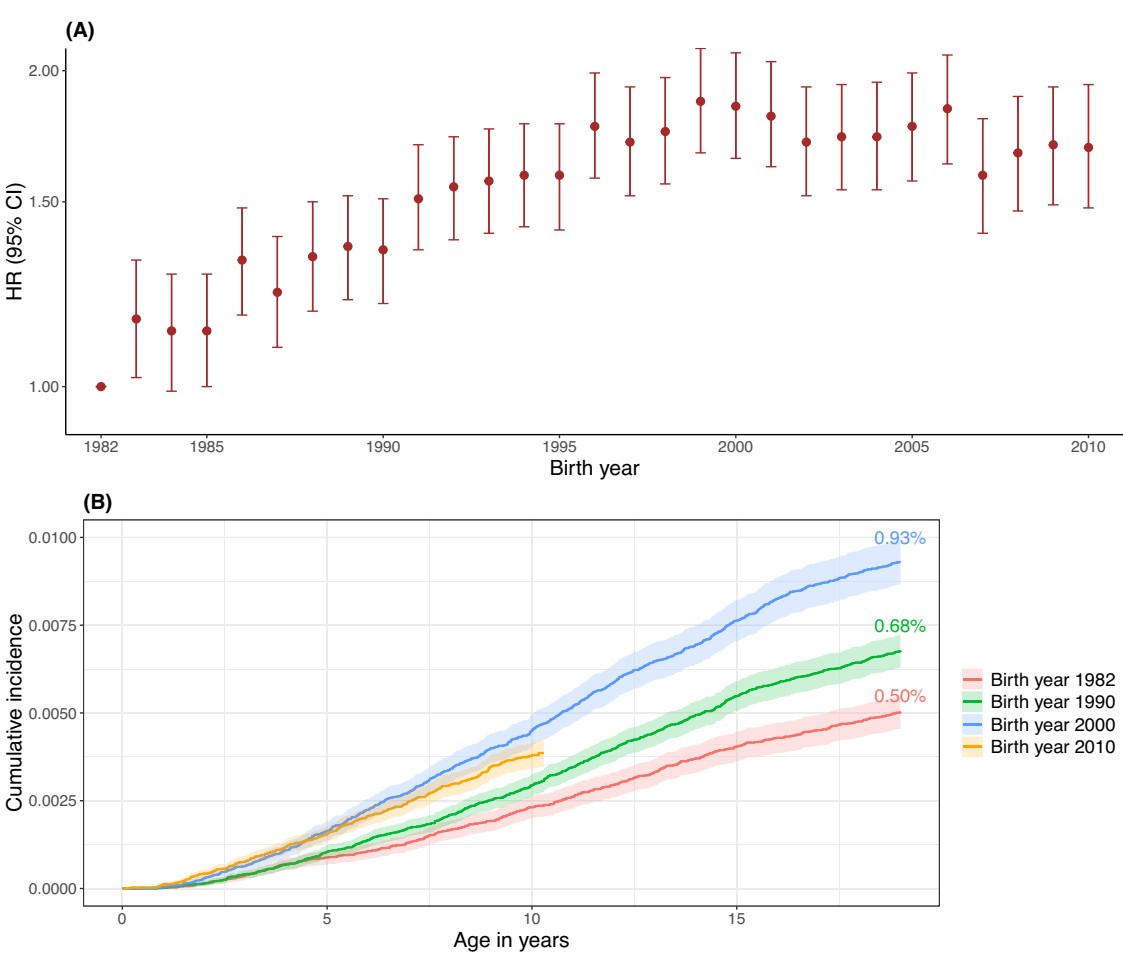

**Fig. 1 | Incidence of type 1 diabetes according to birth year.** HR hazard ratio, CI confidence interval. **A** HR (95% CI) of type 1 diabetes in each birth year (*n* = 2,928,704). The solid circles represent the point estimates of HRs, calculated as exponential of regression coefficients in Cox models, while error bars represent 95% CIs, calculated as exponential of (β coefficients±1.96×standard errors). Cox models were fitted with the birth year of 1982 as the reference group, with attained age as the time scale and with adjustment for sex. Cluster robust sandwich estimator for standard errors were used to correct for the dependence among individuals born by the same mother. Source data (the exact values for HRs and 95% CIs) are provided as a Source Data file. **B** Cumulative incidence of type 1 diabetes before age 19 years in people born in different birth years was estimated using Kaplan Meier curves. The red, green, blue, and orange shaded areas represent the 95% CIs, calculated as cumulative incidence estimates ±1.96×standard errors, for birth year 1982, 1990, 2000, and 2010, respectively.

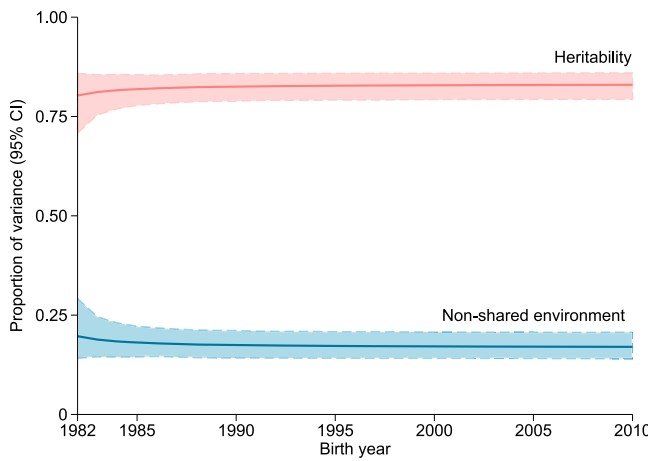

**Fig. 2 | Proportion of variance in type 1 diabetes explained by genetic (heritability) and non-shared environmental factors according to birth year.** CI confidence interval. The solid pink and blue lines represent the point estimates of proportions of type 1 diabetes variance contributed by genetic (heritability) and non-shared environment components in the AE model, respectively, and the shaded areas represent 95% CIs estimated using bootstrapping approach. The AE model was fitted with moderation of birth year on both genetic and environmental components, with adjustment for sex and birth year for the liability threshold. The heritability was 0.83 (95% CI: 0.79, 0.86) and non-nonshared environmental contribution was 0.17 (95% CI: 0.14, 0.21) for the whole birth cohort period (1982–2010). Source data (the exact values for point estimates and 95% CIs) are provided as a Source Data file.

cases per 10,000 children (the increasing incidence from birth year 1982 to 2000) is completely due to changing environmental risk factors (Supplementary Table 6).

### Environmental factors

Maternal smoking, BMI and bacterial infection during pregnancy, gestational age, mode of delivery and low birth weight (<1500 g) were associated with T1D in the cohort analysis, and the associations remained in the sibling analysis for all exposures except maternal BMI (Fig. 4, Supplementary Fig. 4). Among factors inappropriate for sibling analysis, higher maternal age at delivery and lower maternal educational level were associated with higher T1D incidence while there were no associations with death in first- or second- degree relatives (Supplementary Fig. 5) or birth order (Fig. 4).

### Contribution of environmental factors to time trend

Among the seven T1D-related factors identified above, maternal age at delivery and educational levels increased while the prevalence of maternal smoking during pregnancy decreased over time (Supplementary Fig. 6). In the causal mediation analysis, the changing prevalence of maternal smoking during pregnancy and changing maternal age at delivery explained 3.2% and 0.8%, respectively, of the increasing T1D incidence from 1980s to birth year 2000 while the proportion explained by other five factors was almost 0 (Table 1). Based on Swedish nationwide summary-level data, the increasing prevalence of childhood overweight/obesity explained approximately 2–4% and 1–2% of the increasing T1D incidence in boys and girls, respectively (Supplementary Data). Together these factors explained 5–8% of the rise in T1D.

## Discussion

### Main findings

This nationwide study based on 3 million Swedish children revealed that the incidence of T1D almost doubled from 1982 to 2000. Heritability remained consistently high at around 0.80 and so did the heritability for the proposed T1D endotypes proxied by onset-age. Several childhood, pre- and perinatal factors available

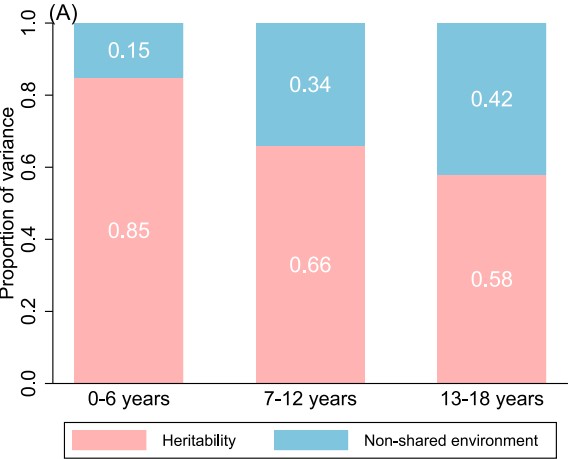

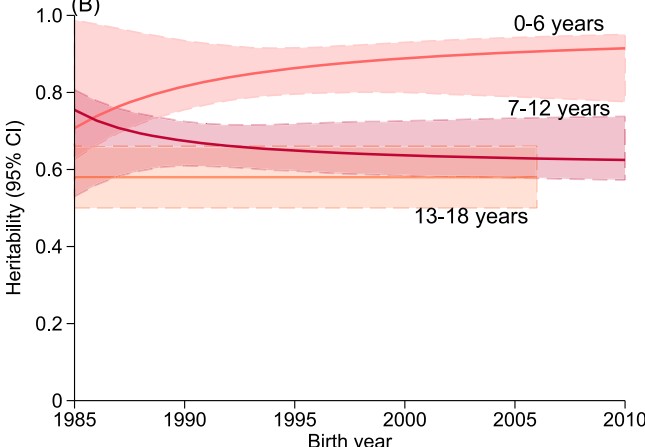

**Fig. 3 | Heritability of type 1 diabetes at different ages (0–6, 7–12, and 13–18 years) during the whole period and by birth year.** CI confidence interval. **A** Proportions of variance in type 1 diabetes at different ages explained by genetic (heritability) and non-shared environmental factors during the whole period. **B** Heritability at different ages in each birth year. Heritability for birth year 1982-1984 was not shown in (**B**) due to the unstable estimates at extremes and the subsequent wide 95% CIs. The solid lines represent the point estimates of T1D heritability at different ages in AE models, and the shaded areas represent 95% CIs estimated using bootstrapping approach. AE models were fitted with moderation of birth year on both genetic and non-shared environmental components. Source data (the exact values for point estimates and 95% CIs) are provided as a Source Data file.

were associated with T1D but changes in their distribution accounted for only a small portion of the increased T1D incidence. Overall, our findings indicate minor changes in the proportions of T1D variance explained by genetic and environmental factors over time.

### Main findings in relation to previous studies

Our findings of a rise in T1D incidence between 1982 and 2000 followed by a plateau confirms previous Swedish studies based on another nationwide register established since 1978[22,23]. A similar rise has been reported worldwide[1,2,24] and a similar levelling off has been observed in some countries including Finland[24,25]. We extend these findings by showing that the increasing trend was observed for different proposed T1D endotypes proxied by age at onset, although the trend for ages 13-18 was unstable after birth year 2005 due to short follow-up and few events.

Heritability reflects genetic expression or penetrance in a given environment[26] and thus provides clues to disease etiology. To

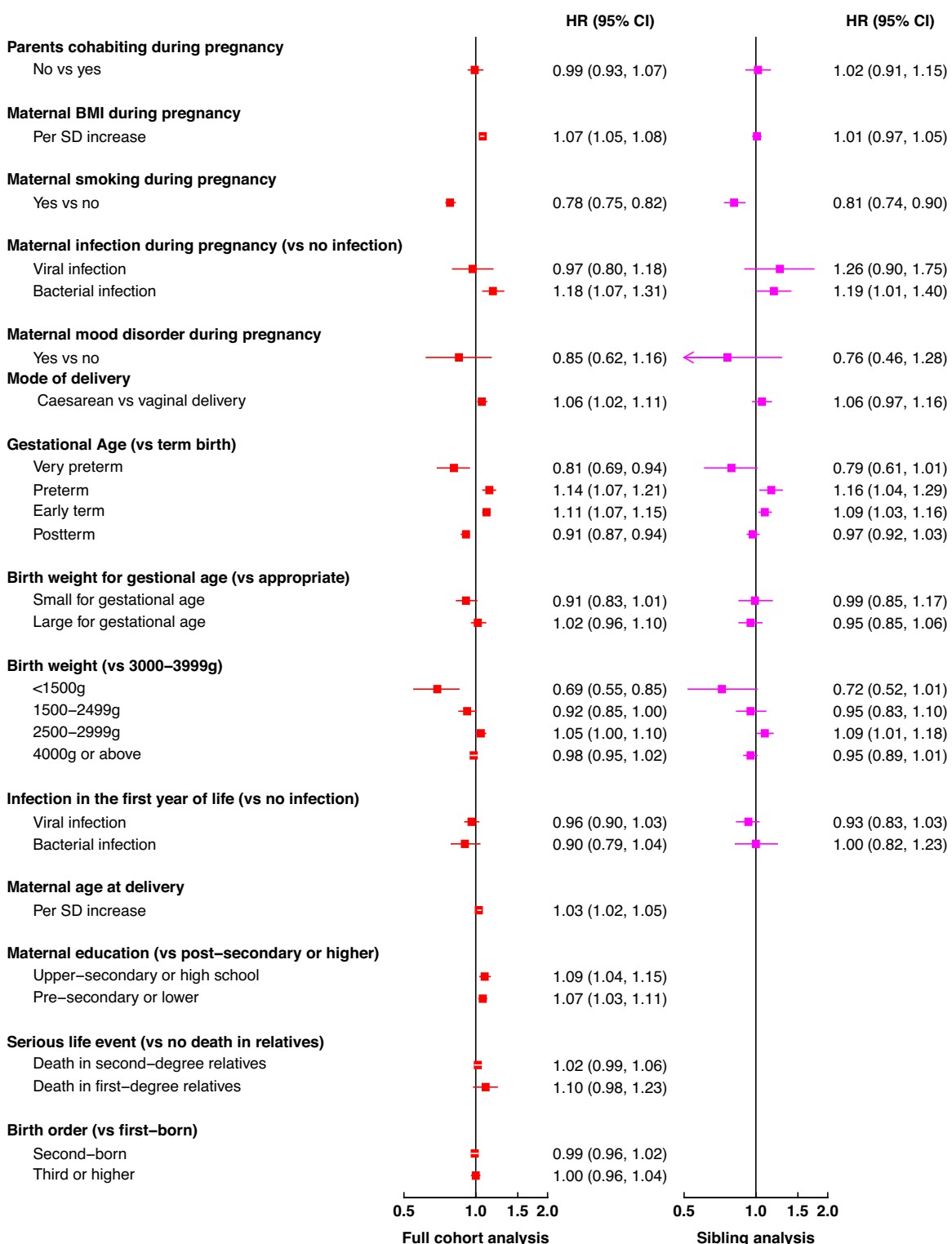

**Fig. 4 | Hazard ratio of type 1 diabetes in relation to different environmental factors in full cohort and sibling analyses.** BMI body mass index, HR hazard ratio, CI confidence interval. The red and pink solid square boxes represented HRs (exponential of regression coefficients) in the full cohort and sibling analyses, respectively, while the error bars represented the 95% CIs. HRs (95% CIs) in the full cohort (n = 2,928,704) were estimated using Cox models with adjustment for birth year, sex, maternal age at delivery, parental education, maternal BMI and smoking during pregnancy, parental country of birth, and parental history of type 1 diabetes, with cluster-robust standard errors to account for correlation among children born by the same mother. Sibling analysis (n = 41,618) was performed using Cox models with the same covariates (except parental country of birth and parental history of type 1 diabetes) as in the full cohort, with stratification by siblings born by the same mother. Sibling analysis was not performed for some factors since within each sibling group, those born later will have outcomes at a later calendar time and will always have a higher maternal age at delivery and birth order, may have higher (or at least not lower) maternal educational level, and are less likely to have experienced a serious life event.

**Table 1 | Proportion of increasing cumulative incidence of type 1 diabetes in children born in 2000 vs 1982^a explained by changing prevalence of different environmental factors**

| Environmental factors | No. of children born in 1982–2000 with available data^a | (A) Increasing incidence of type 1 diabetes from 1982^a to 2000 (per 10,000 children)^b | (B) Increasing incidence due to changing prevalence of the environmental factor (per 10,000 children)^c | (C) Increasing incidence not explained by changing prevalence of the environmental factor (per 10,000 children) | (D) Proportion of increasing incidence explained by changing prevalence of the environmental factor, % |
|---|---|---|---|---|---|
| Maternal age at delivery | 1,919,906 | 40.9 | 0.3 | 40.6 | 0.8 |
| Maternal education | 1,908,532 | 42.4 | -0.2 | 42.5 | -0.4 |
| Maternal smoking during pregnancy | 1,712,843 | 32.6 | 1.1 | 31.6 | 3.2 |
| Maternal bacterial infection during pregnancy | 1,915,368 | 41.6 | 0.0 | 41.6 | 0.0 |
| Mode of delivery | 1,919,906 | 41.4 | 0.2 | 41.3 | 0.4 |
| Gestational age | 1,915,365 | 42.3 | 0.0 | 42.2 | 0.1 |
| Birth weight | 1,912,575 | 41.9 | 0.1 | 41.8 | 0.2 |
| Childhood overweight/obesity (boys)^c | -d | 51.9 | 0.8–1.9^d | 50.0–51.2^d | 1.5–3.6^d |
| Childhood overweight/obesity (girls)^c | -d | 33.1 | 0.2–0.6^d | 32.5–32.9^d | 0.8–1.8^d |

^aIn the analysis of each environmental factor, we excluded children with missing data on the corresponding factor. Therefore, the total increasing incidences (column A) were slightly different for the analyses of different factors. We presented the sample size with available data in 1983–2000 and used 1983 as the reference year in the analysis of maternal smoking during pregnancy due to the high missing rate of maternal smoking during pregnancy in the Medical Birth Register in 1982.
^bThe total increasing incidence is composed of two parts, namely the natural indirect effect mediated through the changing environmental factor (column B) and the natural direct effect not mediated though the changing environmental factor (column C).
^cThe analyses were performed in boys and girls separately due to the sex difference in the prevalence of overweight and obesity.
^dWe did not have individual-level data for childhood overweight/obesity. Columns (B), (C), and (D) were estimated using Swedish Nationwide summary-level data and the strengths of associations between childhood-overweight/obesity. To be prudent, the estimates were made based on different strengths of associations.

understand the driving forces behind the rise in T1D, we estimated time trends in the heritability of T1D. To the best of our knowledge, this has not been done before. We found that the proportions of T1D variance explained by genetic and environmental factors have remained virtually unchanged during the last 30 years. Regarding proposed endotypes of T1D, we find that a larger proportion of T1D variance is explained by genetic factors for younger age at diabetes onset, confirming findings from a heritability study based on ImmunoChip genetic variants[27]. Moreover, no major changes in the heritability of T1D diagnosed at different ages were observed. This suggests that changing environmental and genetic pressure may be influencing the risk of T1D at different ages similarly, although different endotypes may have distinct underlying pathophysiological mechanisms[28,29].

Our simulation results indicated that T1D heritability should have decreased markedly if only environmental factors were driving the increasing T1D incidence. In contrast, heritability in our study was stable at around 0.8. Therefore, it is unlikely that the observed rise in T1D can be solely explained by changes in environmental factors. Our liability threshold model suggests that the combined genetic and environmental loads required to develop T1D might have decreased over time. Thus, genetic factors and changes in environmental factors may jointly have contributed to the rise in T1D. The reasons behind the decreased requirement for the combined loads remain unclear. However, it has been noted that the proportion of children with T1D who have highest-risk HLA genotypes is decreasing while the proportion with moderate-risk HLA genotypes is increasing[16–20]. It is hypothesized that the penetrance of moderate-risk HLA genotypes has increased over time due to increasing environmental exposures[1,16], which implies that genetic loads required to develop T1D might have decreased. Children with a moderately elevated genetic risk, who would not have developed T1D under a less "diabetogenic" environment, may now be developing it due to the more "diabetogenic" environment caused by environmental triggers that have become more prevalent. This can be considered a form of gene-environment interaction and therefore supports a joint contribution from genetic and environmental factors to the rise in T1D.

We investigated a wide range of environmental (childhood, pre- and perinatal) factors and could confirm that maternal smoking during pregnancy[7], maternal infection during pregnancy[30], maternal age at delivery[8], maternal education[31], gestational age and extremely low birth weight[32] are associated with T1D, while our sibling analyses indicate that the association with maternal BMI during pregnancy may be due to familial confounding. Shifts in the prevalence of some of these factors were noted but this could only explain a small proportion of the observed rise in T1D. To the best of our knowledge, ours is the first study to quantify how changes in various environmental factors contribute to the increasing T1D incidence based on individual-level data. Using summary level data, we also estimated that the proportion of increasing incidence explained by childhood overweight/obesity was also small. Our findings are in line with a previous simulation study which reported that none of the known T1D-related environmental factors could explain the rapid increase in T1D incidence[1]. Altogether, changes in the distribution of all the environmental factors we investigated explained less than 10% of the rise in T1D. Our findings indicate that the drivers behind the trend may include other environmental factors that are not investigated here. Such factors are most likely those not shared by siblings such as factors in the first year of life, since shared environmental factors have little contribution to T1D[14]. Breastfeeding is associated with lower T1D risk[9] while antibiotic exposure prenatally and during the first year of life is associated with higher risk[33]. Still, since the prevalence of exclusive breastfeeding increased from 1980s to 2000 and the prevalence of antibiotic use has been declining in Sweden[33], it is unlikely that these factors explain the rise. Dietary factors within the first year of life other than breastfeeding could also play a role, such as late introduction of gluten, fruit and

cow's milk, and high consumption of cow's milk[9]. It is also possible that changes in unknown environmental factors have caused a more "diabetogenic" environment and contributed to the increasing T1D incidence. Future large-scale cohort studies with individual-level data are warranted to identify such factors.

### Strengths and limitations

The main strength is that the heritability estimate by birth year was made possible by the world's largest nationwide multi-generation register, which allowed us to link all siblings born in Sweden since 1982. The second is the inclusion of a nationwide population with linkages to high-quality registers for assessment of T1D and environmental factors and the implementation of a sibling comparison design to minimize familial confounding. Third, we integrated different methods to investigate the reasons for the increasing T1D incidence from different aspects, namely the calculation of T1D heritability over time, the simulation analysis, and the quantification of increasing incidence explained by changing prevalence of available environmental factors. One limitation is that NDR and outpatient data for T1D assessment were unavailable at the start of the study period (1982). However, even if some cases occurring in earlier years were initially missed by the inpatient data, they would eventually be captured by NDR (established since 1996) and outpatient data (available since 2001) before age 19 years. In addition, NDR records year at diabetes diagnosis even for cases diagnosed before 1996 and this helps to more accurately estimate the age at T1D diagnosis. Therefore, we believe that we were able to capture almost all childhood-onset T1D cases during the whole study period and the analysis of T1D incidence and heritability by birth year is unlikely to be biased. Islet autoimmunity and T1D are considered as results of a complex interplay between host genetics and environmental factors[34]. Therefore, the rise in T1D might be partly due to the interaction between genetic factors such as moderate-risk HLA genotypes and emerging environmental factors such as childhood obesity[35,36]. As an example, obesity may induce insulin resistance, which could increase β-cell stress and intensify the autoimmune response in children who are genetically predisposed[37]. Without individual-level data on childhood overweight/obesity or specific genetic data at hand, we cannot assess this hypothesis. Future large-scale cohort studies with comprehensive measurements of genetic and environmental factors are warranted to explore this issue. Enterovirus infections are linked to T1D[34] but we could not differentiate it from infections due to other pathogens. This study was conducted in Sweden and the generalizability of our findings to other settings is uncertain.

In conclusion, our findings indicate that the heritability of T1D remained high and stable in Sweden between 1982–2020 and thus, the proportion of T1D variance explained by environmental factors does not seem to be increasing. This indicates that both environment and genetics contributed to the rise in T1D incidence noted in 1982–2000. The environmental factors we investigated, such as early-life factors and childhood adiposity, played a minor role in explaining the rise of T1D, indicating that other environmental factors are at play. These factors remain to be identified.

## Methods

This research complies with all relevant ethical regulations. This study was approved by the Swedish Ethical Review Authority (2023-01799-02), who also waived the requirement for participant consent.

### Study population

We identified all children born since 1982 from the Swedish Medical Birth Register (MBR), which was established in 1973. The children were linked to their parents and siblings through the Multi-Generation Register (MGR), which includes linkages to parents for individuals born since 1932 and still alive in January of 1961[38]. We excluded stillbirths, neonates who died within the first month of birth, children without identifiable biological parents, or with contradictory sex information between MGR and MBR. Unless stated otherwise, the study population was children born in 1982-2010 ($n$ = 2,928,704) (Supplementary Fig. 7).

### T1D assessment

Participants were followed up for T1D diagnosis in the National Patient Register (NPR) and National Diabetes Register (NDR) from birth until they turned 19 years, age of emigration or death (through the Total Population Register), or age in 2020, whichever came first (Supplementary Method 1). The date of diagnosis was the earliest recording in NPR, NDR, or the first prescription for glucose-lowering drugs in the Swedish National Prescribed Drug Register, whichever came first. We further categorized T1D into potential endotypes according to age at diagnosis (0–6, 7–12, and 13–18 years). T1D in parents was defined as an exclusive diagnosis of T1D in NDR or NPR.

### Environmental factors

We systematically searched for review, meta-analysis, and original articles (Supplementary Method 2) to identify all environmental factors linked to childhood-onset T1D (Supplementary Table 7).

Information on all T1D-associated factors available in the nationwide registers was retrieved (Supplementary Table 7), including birth year and month, sex, gestational age, birth weight, mode of delivery, maternal marital (cohabiting) status during pregnancy, maternal age at delivery, maternal smoking and body mass index (BMI) during pregnancy, parental educational attainment, exposure to serious life events, maternal mood disorder during pregnancy, maternal infections during pregnancy and infections during the first year of life (Supplementary Method 3, Supplementary Table 8). We lacked individual-level data for childhood overweight/obesity, and obtained prevalence data for children born in different birth years from previous Swedish nationwide studies[39] and information on its strengths of associations with T1D from previous reports[40].

### Statistical analysis

**Incidence of T1D by birth year.** We estimated the incidence trend of T1D, overall and by age at diagnosis (ages 0–6, 7–12, and 13–18 years), by estimating the hazard ratio (HR) in each birth year in 1983–2010 in a Cox proportional hazards regression model, with children born in 1982 as the reference group. The model used attained age as time scale, adjusted for sex, and with cluster-robust standard errors to account for the correlation among children born by the same mother. We also estimated the cumulative incidence of T1D before age 19 years by birth year.

**Heritability.** Disease variance can be decomposed into additive genetic variance (A), dominant genetic variance (D), environmental variance shared (C) or non-shared within siblings (E). We recently showed that only additive genetic effect (A) and environmental factors not shared within siblings (E) contribute to the variance of childhood-onset T1D in Sweden, while no evidence was found for the influence of shared environmental factors or dominant genetic effects[14]. Therefore, we used an AE model to calculate heritability which requires one type of relative pairs. Consequently, the analysis of heritability was based on 1,549,357 pairs of full siblings (sharing ~50% of A) born in 1982–2010. All possible full sibling pairs from each family were selected to increase statistical power.

To assess potential changes in the proportions of T1D variance explained by genetic (A) and non-shared environmental factors (E) over birth year, we fitted liability threshold models[10] (AE model). Models were fitted with moderation effects[41,42] of birth year on the variance of genetic component (model 2), environmental component

(model 3), or both (model 1), with adjustment for sex and birth year for the liability threshold (Supplementary Method 4; Supplementary Fig. 2). In each model, we fixed the total variance of T1D at 1 at the median value of birth years (1996) and allowed the total variance to change over birth years[41]. However, we only interpret the proportions of variance explained by A (heritability) and E, since the change in total variance in the model cannot readily be meaningfully interpreted. The models were applied to T1D overall and separately by age at diagnosis (ages 0–6, 7–12, 13–18 years). We also fitted the model without any moderation effect (model 4) and compared the fitting of models 1 to 4 based on likelihood ratio tests (-2LL) and Akaike information criterion (AIC) values. We performed a sensitivity analysis by comparing model fitting based on only one full sibling pair from each family. The best-fitting model was used to estimate the heritability in each birth year as the main results (Supplementary Method 4). The 95% CIs for heritability and non-shared environment contribution to T1D variance were estimated using bootstrapping to account for dependency between sibling pairs. Bootstrapping was performed by randomly drawing as many families (with replacement) as in the original dataset 1000 times, and the middle 95% (2.5% to 97.5%) of the produced distribution of parameters were used as CIs[14].

We performed a simulation study to estimate the expected heritability in a scenario where an increasing cumulative incidence of 40 cases per 10,000 children over time is completely driven by environmental factors (Supplementary Method 5).

**Environmental factors.** We assessed the associations of all potential and available T1D-related environmental factors, with T1D in the full cohort. Estimates of effect sizes were calculated using Cox models (Supplementary Method 6). In addition, we used a sibling comparison design, when appropriate (Supplementary Method 6), with Cox models stratified by sibling groups. Such a design compares outcome hazards among siblings with different exposure status and can control for unmeasured confounding factors (genetic and environmental) shared within families[43]. An HR close to 1 in the sibling analysis indicates that the association observed in the overall cohort is due to confounding factors shared within families.

**Contribution of environmental factors to time trend.** We considered all environmental factors associated with T1D in the full cohort analysis, except those with an HR close to 1 in the sibling analysis, as T1D-related factors (Supplementary Fig. 7). We used causal mediation models adapted into a survival setting[44] to quantify the proportion of increasing cumulative incidence of T1D explained by the changing prevalence of these factors during the birth cohort period when T1D incidence increased. Such models divided the increasing cumulative incidence (1 minus survival function) observed over birth year into a) natural indirect effects mediated through the changing prevalence of each T1D-related environmental factor and b) natural direct effects not mediated through that factor[44]. We estimated the proportion of the increasing cumulative incidence explained by overweight/obesity using summary-level data from previous studies[39,40,45] (Supplementary Method 7).

Calculation of heritability (OpenMx package) and cumulative incidence of T1D, and mediation analysis were performed in R 4.0.4. Excel (Microsoft 365) was used to calculate the proportion of increasing T1D incidence explained by changing prevalence of childhood overweight/obesity. Other analyses were performed using STATA 17.0. All analyses were two-sided, with p < 0.05 as the statistically significant level.

**Reporting summary**
Further information on research design is available in the Nature Portfolio Reporting Summary linked to this article.

## Data availability
Our individual-level data is register-based data that is pseudonymized and thus subject to GDPR and cannot be shared openly. Only metadata is published openly. Underlying data from the registers can be made available upon request to rdo@ki.se for researchers who have relevant ethical approval from the Swedish Ethical Review Authority and after ensuring compliance with relevant legislation and GDPR. The time-frame for responses to requests is 3 months. Metadata of this study are provided in GitHub (https://github.com/Yuxia-Wei/T1D_trend_heritability). They can also be accessed in Zenodo (https://doi.org/10.5281/zenodo.15384742). Source data (the exact values for point estimates and 95% confidence intervals) used to plot Fig. 1(A), Fig. 2, and Fig. 3(B) are provided with this paper. All other data supporting the findings of this study are available in the article and its Supplementary Information files. Source data are provided with this paper.

## Code availability
Codes used in this study have been made publicly available on GitHub (https://github.com/Yuxia-Wei/T1D_trend_heritability) under MIT license[46]. They can also be accessed in Zenodo (https://doi.org/10.5281/zenodo.15384742).

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

## Acknowledgements

This study was supported by the Swedish Research Council (grant number 2022-00811, to S.C.), the Swedish Diabetes Foundation (grant number DIA2022-735, to S.C.), the Swedish Childhood Diabetes Foundation (grant number 2024-0006, to S.C.), and the China Scholarship Council (student number 202006010041, to Y.W.). The sponsors had no role in the study design, data collection, data analysis and interpretation, writing of the report, or the decision to submit the article for publication. We confirm the independence of researchers from funders and that all authors had full access to the data in the study and can take responsibility for the integrity of the data and the accuracy of the data analysis.

## Author contributions

Y.W. and S.C. conceived and designed the study. M.F. acquired and managed data. R.K., Y.W., T.A., and S.L. contributed to methodological issues. R.K. helped with coding. Y.W. analyzed data and drafted the first draft of the manuscript. All authors critically revised the manuscript for important intellectual content and made substantial contributions to the interpretation of data. All authors reviewed and approved the final manuscript. All authors confirm that they have full access to all the data in the study and accept responsibility to submit for publication. Y.W. and S.C. are guarantors of this study.

## Funding

## Competing interests

The authors declare no competing interests.

## Inclusion and ethics in global research

This study is based on participants in Sweden. All authors are local researchers in Sweden and are involved throughout the research process. There are no multi-region collaborations.
