## [Peer review file · Nature Communications]

Stable Heritability of Type 1 Diabetes in a Swedish Nationwide Cohort Study

Corresponding Author: Dr Yuxia Wei

Version 0:

Reviewer comments:

Reviewer #1

(Remarks to the Author)

Wei et al have used a large set of complete population data from Sweden to investigate potential contributors to the increasing time trend in type 1 diabetes that have been seen over the past few decades in multiple countries without a readily acceptable explanation. They estimate that the heritability seems stable over time and that the increasing cumulative incidence over birth cohorts cannot be explained by proposed non-genetic risk factors. While a decreasing heritability over time would have been easier to explain, the observed results are difficult to digest. Nevertheless, this is a novel and relevant contribution, a well performed study and well written manuscript. Limitations include that there is no data on genetic markers, only sibships. A few clarifications regarding interpretation of heritability estimates over time would improve the manuscript, and some other issues should be addressed – see comments below.

COMMENTS

1. Heritability of a binary trait. An inherent problem with this is that the variance of a binary variable depends on the incidence proportion (cumulative incidence) via the variance formula $p(1-p)$, so the variance approaches zero when p approach zero and increases abruptly with increasing cumulative incidence. It is customary to base analyses on liability threshold model. I do not have hands on experience with such analyses but understand from the cited literature that identification requires setting the total variance to unity (possibly within levels of a “moderator”). Could you please help explain what this really mean. In other words, if there is truly an increasing total variance over time and you enforce a constraint that the total is constant, could this lead to (potentially) misleading overall results? (different age-groups may also differ in cumulative incidence, so similar arguments apply, I think). Can we escape the fact that when the cumulative incidence of T1D increased from around 0.003 to around 0.009, the true overall variance of T1D as a binary trait increased with the approximately the same values, which is a 3 fold increase?

2. Plausibility of increased contribution of genes over time to explain increasing T1D occurrence. It is always good to challenge dogmas such as increasing incidence of T1D must be due to environmental factors. The authors both repeat the dogma and discuss increased fertility in women with T1D, as I understand in an attempt to explain their intriguing findings. In my mind, it is not conceivable that the population frequency of susceptibility variants have changed markedly during 30 decades. Even in the proposed scenario that women with T1D have had more offspring over time, the strongest susceptibility variants for T1D (HLA DQ8- and DQ2 haplotypes) are very common in the population. In a setting where the proportion of women with T1D is <1%, I cannot see how even a marked increase in fertility from essentially zero to for instance the current population average in Sweden could have led to a marked change in frequency of susceptibility variants over the 10 decades since the discovery of insulin. (it is commonly believed that if anything, evolutionary selective pressure via the HLA genes have been due to infection susceptibility, even though also this is not well supported by data). Other mechanisms of selection may govern non-HLA susceptibility variants, but the effect size of each of these are also very weak with consequently need for more dramatic changes in population allele frequencies to impact population disease risk. I may be wrong, but I encourage the authors to elaborate a bit more on the potential quantitative impact of increased fertility in women with T1D over time before putting this forward as an argument. (I know others have suggested this before, but as you understand, I remain sceptical.).

Minor comments

a. Title: Consider to clarify that “genetically informed” does not mean there is information about molecular genetic

susceptibility markers, but that there is information about relatedness between study participants – sibships.

b. Abstract/conclusion: consider to add “relative” contribution of environment....

c. Time trends were studied only for cohorts (cohort effects), not period effects. This is likely the most feasible to do but I wonder if there may be limitations to this/whether period effect analyses would give similar or different results in some conceivable scenarios?

d. Methods section, T1D assessment: date of diagnosis first prescription of glucose lowering drug: should this be “insulins” or does it also include non-insulins?

e. Citing literature for the liability threshold model. The details of the models and how it is applied in the current setting may be clear to those who have hands on experience, but the cited ref 16 used twin data and as far as I could see did not explain the model or even cite source literature for their methods. Please clarify why ref 16 is cited, and whether refs 27 and 28 are sufficient refs for your applications. (I guess only ref 28, since the current application is on binary outcome).

(Remarks on code availability)

Reviewer #2

(Remarks to the Author)

Summary

This study investigates the rising incidence of childhood-onset type 1 diabetes (T1D) in Sweden, focusing on the relative contributions of genetic and environmental factors over time. Utilizing a nationwide cohort of 2,928,704 children born between 1982 and 2010 and followed until 2020, the study employed Cox proportional-hazards models to assess incidence trends and liability-threshold modeling to evaluate heritability. Causal mediation was used to assess the role of environmental factors. The findings revealed a near doubling of T1D incidence between 1982 and 2000, with heritability remaining high and stable at approximately 0.83 across this time period. Environmental factors that were available for the authors to analyze collectively explained less than 10% of the increase in incidence. This study provides some insights into the relative contributions of genetic and certain environmental factors to the rising incidence of T1D in Sweden. In particular, it highlights the stability of T1D heritability over time.

Strengths

The use of Sweden’s large registry allowed for linkage of parental, sibling, and individual data, ensuring a high rate of capture for T1D cases during the study period.

This is the first study to analyze heritability trends of T1D over three decades, offering novel insights into the stability of genetic contributions despite rising incidence.

The application of liability-threshold modeling, sibling comparisons, and causal mediation analyses enable some disentanglement of genetic and certain environmental contributions.

Comments and Concerns

The title of the manuscript is a bit confusing. What do the authors mean by a “Genetically Informed Cohort Study”, given that they themselves have listed as a limitation that they do not have genetic data? It would be useful to explain in the manuscript what the authors mean by genetically informed cohort study.

The study did not examine key environmental contributors such as pollutants (e.g., heavy metals, pesticides, or air pollution), gut microbiota alterations, and specific viral infections (e.g., enteroviruses), and dietary factors such as vitamin D, gluten, and cow’s milk exposure, that have been implicated in other studies, limiting the ability to comprehensively evaluate environmental contributions to the rising T1D incidence. While the authors acknowledge that they could not analyze these factors, they should modify their conclusions about environmental factors not explaining the increase. Perhaps it would suffice it to say ‘the environmental factors that were available to analyze...’ or something to that effect.

The lack of individual-level data for childhood overweight/obesity is a limitation and takes away from the other individual level comparisons that investigators are able to make. Because of this, this reviewer recommends not including the analysis of overweight/obesity.

Line 257, the authors state ‘Children with a moderately elevated genetic risk, who would not have developed T1D under a less “diabetogenic” environment, may now be developing it due to the more “diabetogenic” environment caused by environmental triggers that have become more prevalent. This would support a genetic contribution to the rise in T1D.’ This is a well-established hypothesis, but the conclusion that this would represent a genetic contribution seems odd. If the rise of T1D incidence is due to increasing environmental pressure (with the prevalence of the moderately-elevated genetic risk markers remaining the same), that would seem to be an environmental cause (in the context of a gene-environment interaction), not a genetic cause per se.

The authors’ simulation study suggests heritability would decline if environmental factors were the only drivers of the rising T1D incidence over time. The authors raise the point that genetics and environmental factors may jointly be contributing to

the increasing incidence, either through gene-environment interaction or a combined effect (ie, a combination of increasing prevalence of at-risk genotypes and changing environment at the same time). This seems relatively obvious, given that there must be an explanation for the increasing incidence of T1D. An excellent addition to this manuscript would be if they could model this in their simulation study (ie the impact of various examples of gene-environment interaction or a combined effects of genes and environment on incidence). In particular, it would be interesting to model the hypothesis of increasing penetrance of moderate risk HLA genotypes due to increasing environmental pressure (ie what would that look like in terms of heritability over time?).

Throughout the paper, it is the opinion of this reviewer that the authors use slightly too definitive language to describe their findings. For instance, in the discussion section starting at line 224: "Overall, our findings indicate minor changes in the relative importance of genetic and environmental factors on T1D etiology over time." In line 240, the authors state 'Our findings indicate that the relative contributions of genetic and environmental factors to T1D etiology have remained virtually unchanged during the last 30 years.' Without understanding whether gene environment interaction or combined effects are at play, and that many environmental candidates were not tested, it might be prudent to use more conservative language.

The observation that older-onset cases have lower heritability than younger-onset cases is not novel, but the observation that heritability is stable across all age groups is useful because it suggests that changing environmental pressure is influencing all age-groups similarly. Is it possible that there is more misclassification of diabetes in the older age groups (ie type 2 diabetes or type 1.5 diabetes)? Within eFigure 3, there appears to be a change occurring around the year 2005, depicting a lower incidence in the 13-18 age group. What might be responsible for this? Is this a methodologic difference (ie change in the definition of disease/ascertainment of cases over time (ie that may differently distinguish between T1D and T2D) or is it a true decrease in incidence in this one age group? It would be helpful for the authors to address this.

In eFigure 6, there is a notably steep increase in maternal education (post-secondary or higher) starting in 1999. Is this 'real' (and are there other data sources that confirm this?) or is it due to a change in the definition of the education variable or the way it was extracted?

On line 261, the authors state: "A genetic explanation is also partly supported by the fact that women with T1D are more likely to have children nowadays due to improved diabetes care(10)". This reviewer does not see the direct support for this statement on the referenced CDC page (ie ref 10) – although it is possible that the page might have undergone some changes. The authors should cite a paper(s) that directly supports their point. Otherwise, it would seem that adequate T1D care has been available for several decades, so it seems unlikely that more women with T1D are having children due to improvements in care within the timeline of the present study.

(Remarks on code availability)

Reviewer #3

(Remarks to the Author)

(Remarks on code availability)

Reviewer #4

(Remarks to the Author)

This is a study about heritability estimation of type 1 diabetes using a large-scale Swedish national registrar dataset. It investigated an interesting question about whether heritability has changed over the past decades. Heritability estimation is mostly model dependent, and different models would yield varying estimates of heritability. One model in this study suggests the heritability of T1D has been stable over time, although it is unclear whether other models would support the same conclusion. Also, there are a few misleading statements about heritability in the text – specifically, the misunderstanding that it measures the "contribution" of genetic factors to disease risk.

Despite these caveats, this study does look at heritability from an interesting perspective. Its focus on how T1D heritability may or may not change over time is a valuable contribution to the field of genetic research.

Comments for the main text

Lines 59-62: This is unclear. Can you elaborate and give more mathematical evidence to back up it?

Lines 124-126: This is unclear and not necessarily correct. However, the statement "one type of relative pairs is enough for heritability calculation" is probably true. The basic model for heritability is just $P = G + E$, where P (phenotype), G (genetic), and E (environmental) are all random vectors. The purpose of discerning relative relationship is just to constructing the variance-covariance matrix of G (genetic). In theory, whatever relative pairs you are using, as long as the genetic relationship you are referencing is correct, the calculation of heritability would not be affected.

Lines 132-140: Birth year acts as a surrogate variable, essentially capturing trends in other environmental factors over time. The moderation effects of birth year may absorb most of the variation of environmental factors, so the remaining random

effects would be insignificant. AIC and other information criteria help pick the most efficient model that fit the data well. However, good models are not necessarily correct models. Would you also show the results of model 4 with no moderation effects of birth year?

Lines 239-241: This is a misleading statement about heritability. Heritability does not say “contribution” of genetic factors to a phenotype. It only estimates how much of the “variation” of phenotype can be explained by the genetic variation. A better and statistically equivalent definition of heritability is originated from selective breeding (so called the Breeder’s equation). Heritability is the responsiveness of selection.

Lines 241-242: Likewise, this is an incorrect statement. “More heritable” is a not a well-defined biological concept. Again, heritability is about response to selection. It represents how much of a trait is maneuverable by breeders.

Lines 244-247: Same. “Importance of environmental and genetic factors” is vague and not well-defined in biology.

Lines 308-313: The idea of “diabetogenic” environment you mentioned gives a good example of gene-environment interaction. Could you have more discussion about how gene-environment interactions could contribute to the development of complex diseases?

(Remarks on code availability)

Version 1:

Reviewer comments:

Reviewer #1

(Remarks to the Author)

The authors have responded adequately to my comments and revised accordingly

(Remarks on code availability)

Even though I did not see the code I it would be good to share all code used for these analyses to facilitate replication with other similar data sets by others (obviously without sharing sensitive data)

Reviewer #2

(Remarks to the Author)

I have reviewed the revised manuscript and the response. I believe the authors have been largely responsive to our comments and critiques. Regarding the issue of summary level overweight/obesity data, we appreciate the author’s rationale for keeping it in the manuscript, and I suggest that the authors include a statement of the lack of individual level data as a limitation in the limitations section of the Discussion, perhaps directly after the sentence discussing the role of obesity in T1D, ending on line 302. In addition, it would be clearer if the authors included a column in Table 1, just to the right of the column listing the environmental factors, where they list the actual sample size used in the analysis of each factor. This may help clarify the large differences seen in the estimate of increasing incidence of type 1 diabetes (A).

(Remarks on code availability)

Reviewer #3

(Remarks to the Author)

(Remarks on code availability)

Reviewer #4

(Remarks to the Author)

Thank you for revising the article per my comments. My concerns about the study has been addressed well.

(Remarks on code availability)

Response letter

Reviewer #1 (Remarks to the Author)

Comment 1: Wei et al have used a large set of complete population data from Sweden to investigate potential contributors to the increasing time trend in type 1 diabetes that have been seen over the past few decades in multiple countries without a readily acceptable explanation. They estimate that the heritability seems stable over time and that the increasing cumulative incidence over birth cohorts cannot be explained by proposed non-genetic risk factors. While a decreasing heritability over time would have been easier to explain, the observed results are difficult to digest. Nevertheless, this is a novel and relevant contribution, a well performed study and well written manuscript.

Response: We thank the reviewer for the positive comment regarding the novelty of our study. We agree that the finding of stable heritability is surprising. Importantly, the results were consistent across different models (Figure 2, eFigure 4), demonstrating the robustness of the heritability estimates. Furthermore, our simulation analysis indicates that T1D heritability would indeed decrease if the rise in T1D was completely driven by environmental factors. This implies that genetic and environmental factors may have jointly contributed to the increasing incidence of T1D (the 4th paragraph of the **Discussion**, lines 237-243, page 12).

We have expanded on this topic by adding that according to our liability threshold model, the combined genetic and environmental loads required to develop T1D might have decreased over time. The underlying reasons for the decreased requirement on the combined loads are unclear and need to be further explored in future studies. However, it has been suggested that the penetrance of HLA-related genes has increased over time due to increasing environmental exposures, which implies that genetic loads required to develop T1D might have decreased. The lowered genetic loads needed for T1D development under a more “diabetogenic” environment can be considered as a type of gene-environment interaction and therefore reflects both genetic and environmental contributions to the rise in T1D (the 4th paragraph of the **Discussion** section, page 12).

Comment 2: Limitations include that there is no data on genetic markers, only sibships. A few clarifications regarding interpretation of heritability estimates over time would improve the manuscript, and some other issues should be addressed – see comments below.

Response: We thank the reviewer for this suggestion and have clarified the interpretation of heritability over time in the responses to **comment 1** and **comment 4** below and in the 4th paragraph of the **Discussion** section in the manuscript (page 12). The limitation of having no data on genetic markers is mentioned in the “Strengths and limitations” section of the **Discussion** section (lines 302-303, page 14).

Comment 3: Heritability of a binary trait. An inherent problem with this is that the variance of a binary variable depends on the incidence proportion (cumulative incidence) via the variance formula $p(1-p)$, so the variance approaches zero when p approach zero and increases abruptly with increasing cumulative incidence. It is customary to base analyses on liability threshold model. I do not have hands on experience with such analyses but understand from the cited literature that identification requires setting the total variance to unity (possibly within levels of a “moderator”). Could you please help explain what this really mean. In other words, if there is truly an increasing total variance over time and you enforce a constraint that the total is constant, could this lead to (potentially) misleading overall results? (different age-groups may also differ in cumulative incidence, so similar arguments apply, I think). Can we escape the fact that when the cumulative incidence of T1D increased from around 0.003 to around 0.009, the true overall variance of T1D as a binary trait increased with the approximately the same values, which is a 3 fold increase?

Response: Because prevalence (or cumulative incidence) and absolute variance are perfectly dependent, there is no way to separate prevalence changes from variance changes without specific assumptions. As customary, we have used the assumption that prevalence may be varying while the variance is fixed, using the standard heritability model in human traits. As the reviewer highlights, in this model we analyzed heritability based on the liability scale, which assumes an un-observed normally distributed liability for the disease to exist.¹ The disease becomes manifest when an individual exceeds a threshold on this assumed disease liability, and this threshold is estimated from the data.¹ The liability threshold model is the standard way of estimating heritability and may be seen as more realistic, and/or with simpler assumptions, compared to an “absolute variance model”. We have added the reference on liability threshold model¹ to the 3rd paragraph of the “**Statistical analysis**” section (reference 10 in the revised manuscript).

We are sorry for causing confusion when describing methods of heritability calculation. In our statistical models, the variance is not constant over the moderator (i.e., birth year). In modelling, we fixed the total variance at 1 in the birth year 1996 to ensure identification and allowed the total variance in other birth years to change.² However, we only interpret the proportions of variance explained by A and E, and we do not interpret the modelled changes in variance as meaningful. In other words, as per custom, we did not assume that variance changes in the model were informative about changes in the “real” variance, since the “real” variance cannot be inferred from the statistical model – i.e., we could have had a model with fixed variance to draw the same conclusions, but chose the current set-up due to it being more practical. We have added clarification to the 3rd paragraph of the “**Statistical analysis**” section (lines 122-126, page 7). We have also removed the originally cited literature³ (original reference 16) since it did not estimate heritability over time in exactly the same way as ours. More details about the method of estimating heritability over time are provided in **eMethod 4** in the supplementary material.

Furthermore, if the variance increased over time while we assumed it was fixed, it would still be valid to infer the proportions of variance explained by A and E. So, if the variance contributing to the liability of T1D is increasing, the changing heritability would reflect the

source of this increase, A or E. Note again that, unless we make other assumptions, we cannot infer from the data if the “real” variance is changing and/or the threshold is changing over moderator values (i.e., birth years), since the prevalence and the variance is perfectly dependent for binary traits and no more information can be extracted. Hence, we have adopted the standard approach, and followed the common modelling choices, referenced in the main text and supplemental material.

Comment 4: Plausibility of increased contribution of genes over time to explain increasing T1D occurrence. It is always good to challenge dogmas such as increasing incidence of T1D must be due to environmental factors. The authors both repeat the dogma and discuss increased fertility in women with T1D, as I understand in an attempt to explain their intriguing findings. In my mind, it is not conceivable that the population frequency of susceptibility variants have changed markedly during 30 decades. Even in the proposed scenario that women with T1D have had more offspring over time, the strongest susceptibility variants for T1D (HLA DQ8- and DQ2 haplotypes) are very common in the population. In a setting where the proportion of women with T1D is <1%, I cannot see how even a marked increase in fertility from essentially zero to for instance the current population average in Sweden could have led to a marked change in frequency of susceptibility variants over the 10 decades since the discovery of insulin. (it is commonly believed that if anything, evolutionary selective pressure via the HLA genes have been due to infection susceptibility, even though also this is not well supported by data). Other mechanisms of selection may govern non-HLA susceptibility variants, but the effect size of each of these are also very weak with consequently need for more dramatic changes in population allele frequencies to impact population disease risk. I may be wrong, but I encourage the authors to elaborate a bit more on the potential quantitative impact of increased fertility in women with T1D over time before putting this forward as an argument. (I know others have suggested this before, but as you understand, I remain sceptical.).

Response: We thank the reviewer for this comment. In our population, <0.5% of children have maternal T1D, and we agree with the reviewer that with such a small proportion of T1D in mothers, it is unlikely that increasing fertility of mothers with T1D has contributed largely to the increasing T1D incidence. We have therefore removed the corresponding sentence (“A genetic explanation is also partly supported by the fact that women with T1D are more likely to have children nowadays due to improved diabetes care.”) from the 4th paragraph of the **Discussion** section in the manuscript (page 12). We have also removed corresponding sentences (“A rise in the proportion of genetically susceptible individuals is also possible. This could occur if women with T1D became more likely to have children towards the end of the 20th century, following major advances in diabetes care which reduced the risks of the mother and infant. Data from the US support such a development.”) from the 2nd paragraph of the **Introduction** section (page 3).

In line with the comment by the reviewer (see also response to **Comment 1**) we have extended the discussion on how genetic and environmental factors may have jointly contributed to a rise in T1D (4th paragraph of the **Discussion** section, page 12).

Minor comments

Comment 5: Title: Consider to clarify that “genetically informed” does not mean there is information about molecular genetic susceptibility markers, but that there is information about relatedness between study participants – sibships.

Response: We thank the reviewer for pointing this out. By “genetically informed”, we mean that based on the genetic relatedness between siblings, we are able to quantify the proportion of T1D variance contributed by genetic (heritability) and environmental factors. To avoid causing confusion, we have removed the words of “genetically informed” from the title.

Comment 6. Abstract/conclusion: consider to add “relative” contribution of environment....

Response: We thank the reviewer for this suggestion, but we have moved the corresponding sentence (“The contribution of environmental factors to T1D does not increase over time.”) from the **Abstract** to meet the journal requirement on the upper limit of word count (150 words).

Comment 7. Time trends were studied only for cohorts (cohort effects), not period effects. This is likely the most feasible to do but I wonder if there may be limitations to this/whether period effect analyses would give similar or different results in some conceivable scenarios?

Response: This is an interesting idea. While we realize that there may be differences in conclusions based on period and on cohort, we also recognize that age effects exists and these may vary over time. So, we are not certain whether focusing on period effects would be helpful in terms of solving the age-period-cohort “problem”. We believe that having a fixed maximum follow-up age (19 years), as well as testing lower ages (similar trend results found when stopping follow-up at 6 years), provides a suitable balance between age, period, and cohort. For practical reasons we have chosen to model and present time-trends in terms of cohort effects. A future direction could be to try to discriminate between period and cohort, while making suitable assumptions about age-of-onset, but we don’t see this as within the scope of the current study.

Comment 8. Methods section, T1D assessment: date of diagnosis first prescription of glucose lowering drug: should this be “insulins” or does it also include non-insulins?

Response: We did not restrict glucose-lowering drugs to insulin. Instead, we used diagnostic codes in high-quality nationwide registers (the National Patient and Diabetes Registers) to ensure the correct classification of diabetes types. The positive predictive value of T1D diagnosed at age 30 or younger is as high as 97% for cases identified from the National Diabetes Register⁴ and 95% for cases identified from the National Patient Register.⁵ In addition, we restricted our analysis to childhood-onset T1D (diagnosed before age 19), further ensuring correct classification of T1D.

Comment 9. Citing literature for the liability threshold model. The details of the models and how it is applied in the current setting may be clear to those who have hands on experience, but the cited ref 16 used twin data and as far as I could see did not explain the model or even cite source literature for their methods. Please clarify why ref 16 is cited, and whether refs 27 and 28 are sufficient refs for your applications. (I guess only ref 28, since the current application is on binary outcome).(Remarks on code availability)

Response: We cited the original reference 16 since that study also estimated heritability based on the liability threshold scale.³ However, we have replaced this reference with another one¹ (reference 10 in the revised manuscript) which explains in more details what a liability threshold scale in heritability studies means. Reference 10 in the revised manuscript also used twin data, but we believe that reference 10 is relevant since the liability threshold model applies to different types of relative pairs.

Regarding the source literature for estimating heritability over birth year (the moderator), we used the approach that reference 25 in the revised manuscript (a methodological article) has developed: “when using binary data, constraining the total variance to unity for a given value of the moderator” to ensure identification within the context of a liability threshold model.² The approach for binary outcomes in reference 25 was derived from the Purcell’s approach (reference 26)⁶ for continuous variables. We therefore cited both references 25 and 26 and believe that they are enough for our applications. We are also happy to share the codes for the calculation with reviewers if we can submit codes without sharing the original dataset, as our individual-level data is subject to General Data Protection Regulation (GDPR) and cannot be shared openly.

Reviewer #2 (Remarks to the Author):

Summary

This study investigates the rising incidence of childhood-onset type 1 diabetes (T1D) in Sweden, focusing on the relative contributions of genetic and environmental factors over time. Utilizing a nationwide cohort of 2,928,704 children born between 1982 and 2010 and followed until 2020, the study employed Cox proportional-hazards models to assess incidence trends and liability-threshold modeling to evaluate heritability. Causal mediation was used to assess the role of environmental factors. The findings revealed a near doubling of T1D incidence between 1982 and 2000, with heritability remaining high and stable at approximately 0.83 across this time period. Environmental factors that were available for the authors to analyze collectively explained less than 10% of the increase in incidence. This study provides some insights into the relative contributions of genetic and certain environmental factors to the rising incidence of T1D in Sweden. In particular, it highlights the stability of T1D heritability over time.

Comment 1:

Strengths

The use of Sweden’s large registry allowed for linkage of parental, sibling, and individual data, ensuring a high rate of capture for T1D cases during the study period. This is the first study to analyze heritability trends of T1D over three decades, offering novel insights into the stability of genetic contributions despite rising incidence.

The application of liability-threshold modeling, sibling comparisons, and causal mediation analyses enable some disentanglement of genetic and certain environmental contributions.

Response: We thank the reviewer very much for the positive comments.

Comment 2: The title of the manuscript is a bit confusing. What do the authors mean by a “Genetically Informed Cohort Study”, given that they themselves have listed as a limitation that they do not have genetic data? It would be useful to explain in the manuscript what the authors mean by genetically informed cohort study.

Response: We are sorry for causing the confusion. By “genetically informed”, we mean the genetic relatedness between siblings, based on which we are able to quantify the proportion of T1D variance contributed by genetic (heritability) and environmental factors. To avoid causing confusion, we have removed the words of “genetically informed” from the title.

Comment 3: The study did not examine key environmental contributors such as pollutants (e.g., heavy metals, pesticides, or air pollution), gut microbiota alterations, and specific viral infections (e.g., enteroviruses), and dietary factors such as vitamin D, gluten, and cow’s milk exposure, that have been implicated in other studies, limiting the ability to comprehensively evaluate environmental contributions to the rising T1D incidence. While the authors acknowledge that they could not analyze these factors, they should modify their conclusions about environmental factors not explaining the increase. Perhaps it would suffice it to say ‘the environmental factors that were available to analyze...’ or something to that effect.

Response: We thank the reviewer for this suggestion and have revised the conclusion in the Abstract section that to state that “The available environmental factors are not the major contributors to the rise in type 1 diabetes in Sweden”. We have also added the word of “available” to the 1st paragraph (line 214, page 11) and 6th paragraph (line 289, page 14) of the **Discussion** section.

Comment 4: The lack of individual-level data for childhood overweight/obesity is a limitation and takes away from the other individual level comparisons that investigators are able to make. Because of this, this reviewer recommends not including the analysis of overweight/obesity.

Response: We acknowledge the limitation of not having individual-level data on childhood overweight/obesity and instead relying on national prevalence data for estimating its contribution. However, we believe it is valuable to retain this analysis, as childhood overweight/obesity is one of the few environmental factors confirmed to increase the risk of T1D⁷ and has risen dramatically over time.⁸ While we recognize that the calculation approach is not perfect, our results suggest that childhood overweight/obesity is unlikely to be a major driver of the increase in T1D. We believe this finding is relevant to readers who may otherwise question the potential role of this factor. That said, we remain open to removing this analysis if the reviewer feels strongly about it.

Comment 5: Line 257, the authors state ‘Children with a moderately elevated genetic risk, who would not have developed T1D under a less “diabetogenic” environment, may now be developing it due to the more “diabetogenic” environment caused by environmental triggers

that have become more prevalent. This would support a genetic contribution to the rise in T1D.’ This is a well-established hypothesis, but the conclusion that this would represent a genetic contribution seems odd. If the rise of T1D incidence is due to increasing environmental pressure (with the prevalence of the moderately-elevated genetic risk markers remaining the same), that would seem to be an environmental cause (in the context of a gene-environment interaction), not a genetic cause per se.

Response: We thank the reviewer for pointing out that it is inappropriate to conclude that it reflects the contribution of genetic factors per se if children with a moderately elevated genetic risk develop T1D in a more diabetogenic environment. This can be considered as gene-environment interaction and therefore indicate that genetic and environmental factors jointly contributed to the rise in T1D. We have revised and extended this topic in the 4th paragraph of the **Discussion** section (lines 249-253, page 12).

Comment 6: The authors’ simulation study suggests heritability would decline if environmental factors were the only drivers of the rising T1D incidence over time. The authors raise the point that genetics and environmental factors may jointly be contributing to the increasing incidence, either through gene-environment interaction or a combined effect (ie, a combination of increasing prevalence of at-risk genotypes and changing environment at the same time). This seems relatively obvious, given that there must be an explanation for the increasing incidence of T1D. An excellent addition to this manuscript would be if they could model this in their simulation study (ie the impact of various examples of gene-environment interaction or a combined effects of genes and environment on incidence). In particular, it would be interesting to model the hypothesis of increasing penetrance of moderate risk HLA genotypes due to increasing environmental pressure (ie what would that look like in terms of heritability over time?).

Response: We appreciate the reviewer’s suggestion to model the hypothesis of increasing penetrance of moderate-risk HLA genotypes due to rising environmental exposures. However, the liability threshold model we use—the standard approach for heritability calculations—does not focus on specific genetic loci like HLA genotypes. While this is an interesting avenue for investigation, addressing it would require individual-level genomic and environmental data. As such, we did not conduct these simulation studies but consider this an important question for future research.

Comment 7: Throughout the paper, it is the opinion of this reviewer that the authors use slightly too definitive language to describe their findings. For instance, in the discussion section starting at line 224: “Overall, our findings indicate minor changes in the relative importance of genetic and environmental factors on T1D etiology over time.” In line 240, the authors state ‘Our findings indicate that the relative contributions of genetic and environmental factors to T1D etiology have remained virtually unchanged during the last 30 years.’ Without understanding whether gene environment interaction or combined effects are at play, and that many environmental candidates were not tested, it might be prudent to use more conservative language.

Response: We thank the reviewer for this suggestion. Heritability is the proportion of disease variance explained by genetic factors. Therefore, the direct interpretation of the stable T1D heritability over time is that the proportion of T1D variance explained by genetic factors has remained stable over time and that the proportion of T1D variance explained by environmental factors has also remained stable. We have revised the corresponding sentences in the 1st and 3rd paragraphs of the **Discussion** section and now the new sentences are “Overall, our findings indicate minor changes in the proportions of T1D variance explained by genetic and environmental factors over time” (lines 215-217, page 11) and “the proportions of T1D variance explained by genetic and environmental factors have remained virtually unchanged during the last 30 years.” (lines 229-230, page 11).

Comment 8: The observation that older-onset cases have lower heritability than younger-onset cases is not novel, but the observation that heritability is stable across all age groups is useful because it suggests that changing environmental pressure is influencing all age-groups similarly. Is it possible that there is more misclassification of diabetes in the older age groups (ie type 2 diabetes or type 1.5 diabetes)? Within eFigure 3, there appears to be a change occurring around the year 2005, depicting a lower incidence in the 13-18 age group. What might be responsible for this? Is this a methodologic difference (ie change in the definition of disease/ascertainment of cases over time (ie that may differently distinguish between T1D and T2D) or is it a true decrease in incidence in this one age group? It would be helpful for the authors to address this.

Response: We thank the reviewer for the inspiration. We have added to the 3rd paragraph of the **Discussion** section (lines 234-236, page 12) that the stable heritability across age groups indicates that the changing environmental and genetic pressure may be influencing all age groups similarly.

We appreciate the reviewer’s observation regarding the slightly different trend in the 13-18 age group around the birth year 2005. This difference is unlikely to be due to misclassification of diabetes in this age group, as it appears specifically from 2005 onward rather than across the entire birth cohort period. The incidence of T1D at ages 13-18 was calculated based on children who were free of T1D before age 13, with follow-up beginning at age 13. For children born in 2005, this follow-up period was limited to a maximum of two years (2018-2020), resulting in a small number of T1D cases and, consequently, unstable point estimates. We have incorporated this discussion into the 2nd paragraph of the Discussion section (lines

224-225, page 11) and added a corresponding note to the footnote of **eFigure 3** in the supplementary material.

Comment 9: In eFigure 6, there is a notably steep increase in maternal education (post-secondary or higher) starting in 1999. Is this ‘real’ (and are there other data sources that confirm this?) or is it due to a change in the definition of the education variable or the way it was extracted?

Response: The educational level data were obtained from nationwide registers and align with trends reported by Statistics Sweden, which show a significant rise in overall educational attainment in Sweden since 2000 (<https://www.scb.se/hitta-statistik/sverige-i-siffror/utbildning-jobb-och-pengar/utbildningsnivan-i-sverige/>). This suggests that the observed increase in maternal education (post-secondary or higher; **eFigure 7** in the revised supplementary material) is a real trend rather than a result of changes in variable definition or data extraction.

Comment 10: On line 261, the authors state: “A genetic explanation is also partly supported by the fact that women with T1D are more likely to have children nowadays due to improved diabetes care(10)”. This reviewer does not see the direct support for this statement on the referenced CDC page (ie ref 10) – although it is possible that the page might have undergone some changes. The authors should cite a paper(s) that directly supports their point. Otherwise, it would seem that adequate T1D care has been available for several decades, so it seems unlikely that more women with T1D are having children due to improvements in care within the timeline of the present study.

Response: We thank the reviewer for spotting this. The webpage has been updated and no longer contains information on fertility of women with T1D. Following the comment by the reviewer (and reviewer 1) we have removed the mentioning of fertility of women with T1D from the manuscript (the 2nd paragraph of **Introduction** and 4th paragraph of the **Discussion** section).

Reviewer #3 (Remarks to the Author):

Reviewer #4 (Remarks to the Author):

Comment 1: This is a study about heritability estimation of type 1 diabetes using a large-scale Swedish national registrar dataset. It investigated an interesting question about whether heritability has changed over the past decades. Heritability estimation is mostly model dependent, and different models would yield varying estimates of heritability. One model in this study suggests the heritability of T1D has been stable over time, although it is unclear whether other models would support the same conclusion. Also, there are a few misleading statements about heritability in the text – specifically, the misunderstanding that it measures the “contribution” of genetic factors to disease risk.

Despite these caveats, this study does look at heritability from an interesting perspective. Its focus on how T1D heritability may or may not change over time is a valuable contribution to the field of genetic research.

Response: We thank the reviewer for your interest in our study and for the positive comment. In line with the suggestion by the reviewer we have added heritability estimates based on different models to the manuscript (more details are in the response to **Comment 4** below). We have also rephrased some statements about heritability in the **Discussion** according to the reviewer’s suggestion, as shown in the response to **Comments 5-7** below.

Comment 2: Lines 59-62: This is unclear. Can you elaborate and give more mathematical evidence to back up it?

Response: There are two types of mathematical evidence to support the statement that “*If the increasing T1D incidence is mainly driven by environmental factors, the relative contribution of environmental factors to T1D variance would be expected to increase and heritability decrease over time*”.

(1) One is the result of our simulation analysis. In the simulation analysis, we made a copy of the original cohort participants and randomly assigned 0.4% (the extent of T1D incidence increase from 1982 to 2000 in our real-world data) of those who originally did not have T1D to be T1D cases in the simulated cohort. The assignment of 0.4% T1D cases was random. Therefore, the assignment of these random T1D occurrences to individuals is independent of the assignment of T1D occurrence to their siblings. This simulates the scenario where the increasing T1D incidence over time is completely due to factors not shared by siblings. In other words, the increasing T1D incidence is completely driven by environmental factors while genetic factors do not play any role. The heritability was 0.59 in the simulated cohort. The results therefore support the statement in lines 59-62 (lines 46-49, page 3 in the revised manuscript). More details about the simulation analysis are provided in **eMethod 5** and **eTable 8** in the supplementary material.

(2) The other type of evidence is the formulas for heritability calculation. Under the assumption that contribution to variance from genetics (A) is constant, while variance contribution from environment (E) can differ, we can write: For children born in 1982, $heritability_{1982} = A / (A + E_{1982})$; for children born in 2000, $heritability_{2000} = A / (A + E_{2000})$. If the increasing T1D incidence is mainly driven by variance from environmental factors, $E_{2000} > E_{1982}$ and therefore $heritability_{2000} < heritability_{1982}$.

Comment 3: Lines 124-126: This is unclear and not necessarily correct. However, the statement “one type of relative pairs is enough for heritability calculation” is probably true. The basic model for heritability is just $P = G + E$, where P (phenotype), G (genetic), and E (environmental) are all random vectors. The purpose of discerning relative relationship is just to constructing the variance-covariance matrix of G (genetic). In theory, whatever relative pairs you are using, as long as the genetic relationship you are referencing is correct, the calculation of heritability would not be affected.

Response: We agree with the reviewer that the basic model for heritability calculation is P (phenotypic variance) = G (genetic variance) + E (environmental variance). However, the genetic variance might involve both additive genetic effect (A) and dominant genetic effect (D), and the environmental variance might involve variance contributed by environmental factors shared by siblings (C) and variance contributed by environmental factors not shared by siblings (E). The more parameters that need to be estimated, the more types of relative pairs are required.⁷ Based on our previous study performed in pairs of full siblings and half-siblings, we found no evidence of contribution from dominant genetic effects or shared environmental component to T1D variance.⁸ We therefore estimated the parameters with only one type of sibling pair in the current study.

We are sorry for the unclear statement in the original lines 124-126. We have added to the 2nd paragraph of the **Statistical analysis** section that disease variance can be decomposed into A , D , C , and E and furthermore, that based on our previous study, we deemed that it was enough for us to estimate T1D heritability based on the AE model with only full sibling pairs (lines 109-116, page 6).

Comment 4 : Lines 132-140: Birth year acts as a surrogate variable, essentially capturing trends in other environmental factors over time. The moderation effects of birth year may absorb most of the variation of environmental factors, so the remaining random effects would be insignificant. AIC and other information criteria help pick the most efficient model that fit the data well. However, good models are not necessarily correct models. Would you also show the results of model 4 with no moderation effects of birth year?

Response: Following the reviewer’s suggestion, we have estimated the time trend of T1D heritability based on other models, including model 2 (with moderation of birth year only on the genetic component) and model 3 (with moderation of birth year only on the environmental component). The heritability estimated based on models 2 and 3 was also stable at around 0.8 over time (**eFigure 4**). This shows the robustness of our findings. Since there is no moderation of birth year on either the genetic component or the environmental component in model 4, the heritability estimated from model 4 is, by definition, the same in each birth year. We have added the description of stable heritability over time in models 2-4 to the 2nd paragraph of the **Results** section (lines 182-183, page 9).

Comment 5: Lines 239-241: This is a misleading statement about heritability. Heritability does not say “contribution” of genetic factors to a phenotype. It only estimates how much of the “variation” of phenotype can be explained by the genetic variation. A better and statistically equivalent definition of heritability is originated from selective breeding (so called the Breeder’s equation). Heritability is the responsiveness of selection.

Response: We have revised the corresponding sentence according to the reviewer’s suggestion and now the new sentence in the 3rd paragraph of the **Discussion** section reads “the proportions of T1D variance explained by genetic and environmental factors have remained virtually unchanged during the last 30 years.” (lines 229-230, page 11)

We have also replaced the sentence of “relative contributions of genetic (A) and non-shared environmental factors (E) to T1D...” with “proportions of T1D variance explained by genetic (A) and non-shared environmental factors (E)” in the 3rd paragraph of the **Statistical analysis** section (line 118, page 7), and replaced “the contribution of environmental factors to T1D etiology” with “the proportion of T1D variance explained by environmental factors” in the **last paragraph of the Discussion section** (lines 309-310, page 15). We have also removed the corresponding sentence (“or whether the relative contribution of environmental factors has increased”) from the 3rd paragraph of the **Introduction section** (page 3).

Comment 6: Lines 241-242: Likewise, this is an incorrect statement. “More heritable” is a not a well-defined biological concept. Again, heritability is about response to selection. It represents how much of a trait is maneuverable by breeders.

Response: According to the reviewer’s question, we have replaced the sentence of “T1D is more heritable with younger age at diabetes onset” with “a larger proportion of T1D variance is explained by genetic factors for younger age at diabetes onset” in the 3rd paragraph of the **Discussion** section (line 231-232, page 11).

Comment 7: Lines 244-247: Same. “Importance of environmental and genetic factors” is vague and not well-defined in biology.

Response: As suggested by the reviewer, we have removed the sentence of “the relative importance of environmental and genetic factors have also stayed constant in the development of the suggested endotypes” from the 3rd paragraph of the **Discussion** section (page 12).

Similarly, we have replaced the sentence of “minor changes in the relative importance of genetic and environmental factors on T1D etiology over time” with “minor changes in the proportions of T1D variance explained by genetic and environmental factors over time” in the 1st paragraph of the **Discussion** section (lines 215-217, page 11).

Comment 8: Lines 308-313: The idea of “diabetogenic” environment you mentioned gives a good example of gene-environment interaction. Could you have more discussion about how gene-environment interactions could contribute to the development of complex diseases?

Response: We thank the reviewer for this comment and are now discussing in the 4th paragraph of the **Discussion** section that the potentially decreased genetic load needed for T1D development under more “diabetogenic” environment can be considered as a type of “gene-environment interaction” (page 12). We have also added a sentence to the 6th paragraph

of the **Discussion** section (lines 300-302, page 14) to deepen the discussion on potential interaction between HLA genotypes and obesity stating that “As an example, obesity may induce insulin resistance which could increase β -cell stress and intensify the autoimmune response in children who are genetically predisposed”.

References

- 1 Skov, J. *et al.* Shared etiology of type 1 diabetes and Hashimoto's thyroiditis: a population-based twin study. *Eur J Endocrinol* **186**, 677-685, doi:10.1530/eje-22-0025 (2022).
- 2 Medland, S. E., Neale, M. C., Eaves, L. J. & Neale, B. M. A note on the parameterization of Purcell's G x E model for ordinal and binary data. *Behav Genet* **39**, 220-229, doi:10.1007/s10519-008-9247-7 (2009).
- 3 Taylor, M. J. *et al.* Etiology of Autism Spectrum Disorders and Autistic Traits Over Time. *JAMA Psychiatry* **77**, 936-943, doi:10.1001/jamapsychiatry.2020.0680 (2020).
- 4 Eeg-Olofsson, K. *et al.* Glycemic control and cardiovascular disease in 7,454 patients with type 1 diabetes: an observational study from the Swedish National Diabetes Register (NDR). *Diabetes Care* **33**, 1640-1646, doi:10.2337/dc10-0398 (2010).
- 5 Miao, J., Brismar, K., Nyrén, O., Ugarph-Morawski, A. & Ye, W. Elevated hip fracture risk in type 1 diabetic patients: a population-based cohort study in Sweden. *Diabetes Care* **28**, 2850-2855, doi:10.2337/diacare.28.12.2850 (2005).
- 6 Purcell, S. Variance components models for gene-environment interaction in twin analysis. *Twin Res* **5**, 554-571, doi:10.1375/136905202762342026 (2002).
- 7 Michael C. Neale. & Maes, H. H. M. *Methodology for Genetic Studies of Twins and Families.*, (Dordrecht, Netherlands: Kluwer Academic Publishers, 1992).
- 8 Wei, Y. *et al.* Familial aggregation and heritability of childhood-onset and adult-onset type 1 diabetes: a Swedish register-based cohort study. *Lancet Diabetes Endocrinol*, doi:10.1016/s2213-8587(24)00068-8 (2024).

Response letter

Reviewer #1 (Remarks to the Author):

Comment: The authors have responded adequately to my comments and revised accordingly.

Response: We thank the reviewer for this positive comment.

Reviewer #1 (Remarks on code availability):

Comment: Even though I did not see the code it would be good to share all code used for these analyses to facilitate replication with other similar data sets by others (obviously without sharing sensitive data)

Response: We thank the reviewer for this suggestion and have shared our codes in GitHub with the following link: https://github.com/Yuxia-Wei/T1D_trend_heritability). They can also be accessed in Zenodo (DOI: 10.5281/zenodo.15384742).

Reviewer #2 (Remarks to the Author):

Comment: I have reviewed the revised manuscript and the response. I believe the authors have been largely responsive to our comments and critiques. Regarding the issue of summary level overweight/obesity data, we appreciate the author's rationale for keeping it in the manuscript, and I suggest that the authors include a statement of the lack of individual level data as a limitation in the limitations section of the Discussion, perhaps directly after the sentence discussing the role of obesity in T1D, ending on line 302. In addition, it would be clearer if the authors included a column in Table 1, just to the right of the column listing the environmental factors, where they list the actual sample size used in the analysis of each factor. This may help clarify the large differences seen in the estimate of increasing incidence of type 1 diabetes (A).

Response: We thank the reviewer for this suggestion. We have added the limitation of lacking individual data on childhood overweight/obesity to the “Strengths and limitations” section (lines 210-211, page 10) in the Discussion according to the reviewer’s suggestion. We have also added a column to **Table 1** to present the number of children born in 1982-2000 without missing data for corresponding environmental factors.